# Association of the Maternal Gut Microbiota/Metabolome with Cord Blood CCL17

**DOI:** 10.3390/nu13082837

**Published:** 2021-08-18

**Authors:** Hiromi Tanabe, Kenichi Sakurai, Yumiko Nakanishi, Tamotsu Kato, Yohei Kawasaki, Taiji Nakano, Fumiya Yamaide, Naoko Taguchi-Atarashi, Yuki Shiko, Ikumi Takashima, Masahiro Watanabe, Shingo Ochiai, Hiroshi Ohno, Hideoki Fukuoka, Naoki Shimojo, Chisato Mori

**Affiliations:** 1Center for Preventive Medical Sciences, Chiba University, Chiba 263-8522, Japan; tanabe-h@ncchd.go.jp (H.T.); sakuraik@faculty.chiba-u.jp (K.S.); m_watanabe@chiba-u.jp (M.W.); hideoki.fukuoka@gmail.com (H.F.); shimojo@faculty.chiba-u.jp (N.S.); 2Department of Allergy and Clinical Immunology, National Research Institute for Child Health and Development, Tokyo 157-8535, Japan; 3Laboratory for Intestinal Ecosystem, RIKEN Center for Integrative Medical Sciences, Kanagawa 230-0045, Japan; yumiko.sato.kj@riken.jp (Y.N.); tamotsu.kato@riken.jp (T.K.); naoko.atarashi@riken.jp (N.T.-A.); hiroshi.ohno@riken.jp (H.O.); 4Immunobiology Laboratory, Graduate School of Medical Life Science, Yokohama City University, Kanagawa 230-0045, Japan; 5Faculty of Nursing, Japanese Red Cross College of Nursing, Tokyo 150-0012, Japan; y-kawasaki@redcross.ac.jp; 6Biostatistics Section, Clinical Research Center, Chiba University Hospital, Chiba 260-8677, Japan; shiko_yuki@chiba-u.jp; 7Department of Pediatrics, Graduate School of Medicine, Chiba University, Chiba 260-8670, Japan; t-nakano@chiba-u.jp (T.N.); fyamaide@chiba-u.jp (F.Y.); 8Data Science Office, Clinical Research Promotion Center, University of Tokyo Hospital, Tokyo 113-8655, Japan; takashima193@g.ecc.u-tokyo.ac.jp; 9Department of Bioenvironmental Medicine, Graduate School of Medicine, Chiba University, Chiba 260-8670, Japan; shingoochiai@gmail.com; 10Department of Progressive DOHaD Research, Fukushima Medical University, Fukushima 960-1295, Japan

**Keywords:** cord blood, CCL17/TARC, maternal gut microbiota, short-chain fatty acids

## Abstract

Chemokine (C-C motif) ligand 17 (CCL17) is a pro-allergic factor: high CCL17 levels in cord blood (CB) precede later allergic predisposition. Short-chain fatty acid (SCFA) treatment during pregnancy has been shown to protect mouse pups against allergic diseases. The maternal microbial metabolome during pregnancy may affect fetal allergic immune responses. We therefore examined the associations between CB CCL17 and gut SCFA levels in healthy pregnant Japanese women. CB CCL17 serum levels at birth, and maternal non-specific IgE levels in maternal sera at 32 weeks of gestation were measured. Maternal stool samples were collected at 12 (*n* = 59) and 32 (*n* = 58) weeks of gestation for gut microbiota analysis, based on barcoded 16S rRNA sequencing and metabolite levels. The CB CCL17 levels correlated negatively with butyrate concentrations and positively with isobutyrate at 12 weeks; CB CCL17 correlated positively with valerate and lactate at 32 weeks. Similarly, butyrate levels correlated negatively with maternal non-specific IgE levels, whereas the lactate concentration correlated positively with IgE levels. At 32 weeks, the Shannon diversity index (SDI) of Firmicutes and Proteobacteria correlated negatively with CB CCL17 levels, while those of the total microbiota correlated positively with the CB CCL17 levels. These metabolites may alter fetal immune responses. This study provides the first link between maternal metabolites during pregnancy and the risk of allergic diseases in human offspring.

## 1. Introduction

Recent studies have revealed that allergies in infants begin *in utero*, and are influenced by maternal environmental factors during the pregnancy [1]; notably, a healthy microbiome during the prenatal period appears to be especially important in reducing the risks of asthma and allergic diseases [2]. Balanced and diverse microbial exposure in early life is essential for proper immune development, in terms of both protective anti-microbial and regulatory immune responses to environmental antigens, and regulation of these responses to prevent the excessive reactions that manifest as atopy and asthma [2]. Maternal contact with farm animals and cats that exposes the mother to several microbial populations, during pregnancy, exert distinct effects on regulatory T (Treg) cells, the type-1 helper T cell (Th1)/type-2 helper T cell (Th2) ratio, and FoxP3 demethylation in the offspring [3], thereby influencing the expression of innate immune receptors at birth [4].

The gut microbiota is quantitatively the most important source of microbial stimulation [5], and could be modified by dietary habits [6] or by exercise training [7], and so on. In addition, via the production of short-chain fatty acids (SCFAs) in the metabolome, maternal commensal microbes present during pregnancy may affect the fetal immune system; this may help to explain why the maternal allergy is a substantial risk factor for allergy development in the offspring [5]. Thorburn et al. found that the offspring of pregnant mice treated with SCFA acetates were protected against allergic disease through epigenetic effects: increase H4 acetylation in the FoxP3 promoter region, correlated with increased Tregs in both the mothers and their offspring [8]. Commensal microbes in the gut can induce colonic Treg cells via the production of butyrate, SCFA produced by gut microbes [9]. Maternal allergy, indicated by high total immunoglobulin E (IgE) levels during pregnancy, is a critical risk factor for the development of infantile atopy [10] and high IgE in later childhood [11]. The specific effects of the maternal gut microbiota on the susceptibility of human offspring to allergic disease has been investigated in intervention studies, such as those assessing probiotic and dietary supplementation [12]. However, no data have been presented from human studies relating the maternal microbiota/metabolome composition during pregnancy to an offspring’s risk of developing allergic diseases [13].

High levels of the Th2 cell-attracting chemokine (C-C motif) ligand 17 (CCL17) in cord blood (CB) precedes allergic development later in life [14]. CCL17 is a pro-allergic factor that attracts Th2 cells and promotes Th2 cell differentiation [15]. It is known also to be primarily involved in skin-related allergies [16], and has been strongly correlated with disease severity in infants with atopic dermatitis (AD). Thus, CCL17 plays several roles in allergic inflammation and can serve as an early biomarker to distinguish between different AD phenotypes [17]. Moreover, during pregnancy, CCL17 is produced by the conceptus [18], and higher CB CCL17 levels precede AD in infancy [19] and allergy development in childhood [14]. High CB CCL17 may be associated with the maternal gut microbiota during pregnancy, and this may then affect the epithelial tissues of the offspring at birth, and lead to subsequent allergic symptoms in infancy.

We hypothesized that the maternal gut microbiota/metabolome affects the offspring’s CB CCL17 levels. An advantage of using CB CCL17 levels as a marker to determine the immune response of fetuses is that the CB CCL17 levels can be measured using commercial kits, whereas the non-specific IgE levels in the CB cannot be measured because of their low levels. Previously, we reported that CB CCL17 levels may be associated with the cumulative prevalence of dermatitis in early infancy (DEI) that predisposed the offspring to subsequent allergic symptoms [20]. Moreover, DEI was associated with the maternal intestinal microbiota during pregnancy [20]. However, we have not yet identified the key process; whether maternal intestinal microbiota during pregnancy affect the offspring’s fetal allergic immune response?

The aim of this study was to determine whether the maternal intestinal microbiota and SCFAs during pregnancy influenced the offspring’s fetal allergic immune response. To test the above hypothesis, we tested associations between the maternal fecal microbiota/metabolome during pregnancy and CB CCL17 levels using data from the Chiba Study of Mother and Child Health (C-MACH) [21].

## 2. Materials and Methods

### 2.1. Study Design

We conducted a longitudinal prospective study of three hospital-based birth cohorts, as part of the C-MACH, which was initiated in 2014 in Chiba and Saitama Prefectures near Tokyo, Japan. The study protocol was approved by the ethical boards of the Biomedical Research Ethics Committee of the Graduate School of Medicine, Chiba University (updated ID: 989; application date: 20 September 2019), and written informed consent was obtained from all participants.

Details of the enrolment procedure [21] and maternal gut microbiota analyses have previously been described [20]. Briefly, stool samples were collected from maternal participants in a hospital in the Chiba prefecture associated with C-MACH at 12 (*n* = 59) and 32 (*n* = 58) weeks of gestation for analysis of the gut microbiota and metabolites. In this population, 51 mothers provided stool samples at both time points, but 8 mothers provided stool samples only at 12 weeks and 7 mothers provided stool samples only at 32 weeks of gestation. The maternal gut microbiota was analyzed using the V1-2 variable region (27F-338R) sequencing for 16S rRNA gene sequencing on an Illumina Miseq (Illumina, San Diego, CA, USA). The primers are as follows:

27Fmod:TCGTCGGCAGCGTCAGATGTGTATAAGAGACAGAGRGTTTGATYMTGGCTCAG

338R:GTCTCGTGGGCTCGGAGATGTGTATAAGAGACAGTGCTGCCTCCCGTAGGAGT. Use of this variable region for 16S rRNA gene sequencing has been previously demonstrated [22].

Dual indexes and Illumina sequencing adapters were attached to PCR products using the Nextera XT Index Kit (Illumina). After purification of the amplicon using AMPure XP beads, these samples were quantified using a Quant-iT PicoGreen ds DNA Assay Kit (Life Technologies Japan, Tokyo, Japan). Mixed samples were prepared by pooling approximately equal amounts of PCR amplicons from each sample. A pooled sample library with 20% denatured PhiX spike-in (SeqMatic LLC, Fremont, CA, USA) was sequenced on the Miseq using a 500-cycle kit (Illumina).

Taxonomic assignments and estimation of relative abundance of sequencing data were performed using the analysis pipeline of the QIIME software package [23]. Chimera checking was performed using UCHIME [24]. An operational taxonomic unit (OTU) was defined at 97% similarity. The OTU was assigned a taxonomy based on a comparison with the Greengenes database using RDP classifier [25,26]. The proportions of identified taxa in each sample were summarized and the amount of bacterial diversity was calculated using vegan package in R software.

Details of the measurement of fecal SCFAs have previously been given [27]. Here, we used data to measure succinate, lactate, and SCFA levels.

### 2.2. Exclusions from Fecal Analysis

One participant was excluded from all analyses as she took antibiotics during pregnancy. Several mothers who needed emergency treatment were transferred to other hospitals for advanced medical care, so their CB could not be obtained. Missing CCL17 data for 5 infants at 12 weeks and 3 infants at 32 weeks of gestation let to their exclusion from the correlative analyses between CCL17 levels and each of the SCFAs, the gut microbiota diversity, and relative abundances of taxa in maternal feces. One mother was excluded from analysis of interactions between non-specific IgE and the SCFAs, the gut microbiota diversity and relative abundances of taxa in maternal feces at 32 weeks of gestation, due to missing IgE data. Data were missing for fewer than 11% of all samples.

### 2.3. Measurement of CB CCL17 and Maternal Non-Specific IgE Levels during Pregnancy

The protocol for measurement of CB CCL17 levels has been previously described [20].

Maternal blood samples were collected from all the participants at 32 weeks of gestation. Serum separation and storage were performed using the same method as that for the CB samples. Non-specific IgE levels were measured by SRL, Inc., (Tokyo, Japan) using a Phadia5000 instrument (Thermo Fisher Scientific/Phadia AB, Uppsala, Sweden), according to the manufacturer’s instructions.

### 2.4. Statistical Analyses

*p*-values < 0.05 obtained by a two-tailed test were considered statistically significant. All analyses were performed using the R software package v3.6.3. [28] and SAS software v9.4 for Windows (SAS Institute Inc., Cary, NC, USA).

The normality of the distribution of each dependent variable was assessed using the Shapiro–Wilk normality test. Non-normally distributed variables (i.e., CB CCL17 levels and non-specific IgE levels) were converted to a natural logarithm scale before analysis. The *t*-test was used to assess the correlations between CB CCL17 levels and delivery mode.

Of the nine dominant SCFAs present in maternal feces (detected in >70% of all samples) that could affect CB CCL17 and maternal IgE levels during pregnancy, those with the greatest effect were identified using least-absolute shrinkage and selection operator (adaptive LASSO) [29]. This method was used where sample size was too small for multivariable regression analysis (MRA). It test is a regression analysis method that performs both variable selection and regularization to enhance the accuracy of predictions [30], and penalizes the size of the parameter vector, so that unimportant variables are removed from the model [31], helping to overcome such sample size limitations. We estimated the regression coefficients of independent variables (*β*) [31], as well as the 95% CI and *p*-values, using adaptive LASSO [30]. We used the K-Fold cross-validation method (K = 5) to determine the optimal value for the regularization parameter in the adaptive LASSO [32].

MRA was used to assess the correlations between CB CCL17 levels and several variables, such as the Shannon diversity indices (SDIs) of the total prenatal microbiota and four dominant phyla, and the relative abundances of both the four dominant phyla, and two genera, for which beneficial effects in preventing allergies have been reported [33,34], in maternal feces. The SDI, a measurement of within-sample (alpha-diversity) community diversity, was calculated using the R software package ‘vegan’. The SDI of the total microbiota was calculated using all observed operational taxonomic units (OTUs) after rarefaction. The SDI of each of the four dominant phyla was calculated using all OTUs included in each phylum after the rarefaction. Additionally, MRA was used to analyze the correlations between maternal serum levels of non-specific IgE at 32 weeks of gestation and the SDIs of the four dominant phyla and the total microbiota, and the relative abundances of the four phyla and the two genera in the maternal fecal microbiota. In relation to fecal data and non-specific IgE levels, only those from the same period were used in any given analysis, as the 20-week difference in maturity could not have affected the results.

## 3. Results

### 3.1. Characteristics of Maternal and Newborn Populations

The demographic characteristics (mean ± SD) of the study population are shown in Table 1. No mother was obese (pregravid body mass index (BMI) ≥30). Two neonates of low birth weight (≤2500 g) were included in the population. Delivery mode has been reported to be associated with higher CB CCL17 [35], although, no statistical difference was observed between CB CCL17 levels following spontaneous (10.7 at 12 weeks and 10.9 at 32 weeks of gestation in mean) and caesarean (9.6 at 12 weeks and 9.5 at 32 weeks of gestation in mean) deliveries in this population (*p* = 0.59, spontaneous *n*/caesarean *n* = 48/5 at 12 weeks; *p* = 0.47, spontaneous *n*/caesarean *n* = 47/7 at 32 weeks). Moreover, the gestational ages of the neonates who were delivered by caesarean section (C-section) were 38 to 40 weeks, all analyzed mothers including C-section were healthy, we did not exclude the data from the participants of the C-section for the analyses.

### 3.2. Butyrate in Maternal Feces during Pregnancy Correlated Negatively with CB CCL17, but Isobutyrate, Valerate, and Lactate Correlated Positively with CB CCL17

Butyrate concentrations in maternal feces at 12 weeks of gestation correlated negatively and significantly with CB CCL17 levels. In contrast, isobutyrate concentrations correlated positively and significantly with CB CCL17 levels (Table 2). At 32 weeks of gestation, significant positive correlations between CB CCL17 levels and the concentrations of valerate and lactate were observed (Table 2).

### 3.3. Butyrate in Maternal Feces during Pregnancy Correlated Negatively with Non-Specific IgE, Whereas Lactate Correlated Positively

Butyrate concentrations in the maternal feces at 32 weeks of gestation correlated negatively with non-specific IgE levels in the maternal sera (Table 3), while the lactate concentration correlated positively and significantly with non-specific IgE levels. Butyrate concentrations in the maternal feces at 32 weeks of gestation did not correlate with the SDIs for total microbiota or any of the four dominant phyla (MRA: *p* = 0.73 for total, *p* = 1.00 for Firmicutes, *p* = 0.38 for Proteobacteria, *p* = 0.51 for Actinobacteria, and *p* = 0.74 for Bacteroidetes), or with the relative abundance of taxa (MRA: *p* = 0.21 for Firmicutes, *p* = 0.51 for Proteobacteria, *p* = 0.23 for Actinobacteria, *p* = 0.24 for Bacteroidetes, *p* = 0.51 for *Clostridium*, and *p* = 0.28 for *Lactobacillus*).

### 3.4. Diversity of Maternal Fecal Firmicutes Correlated Negatively with CB CCL17

A recent report has revealed that the human gut microbiota mainly consists of four dominant phyla: Firmicutes, Bacteroidetes, Proteobacteria, and Actinobacteria, and that in total, these comprise more than 95% of total gut microbes [36]. Thus, we focused mainly on these four phyla.

We compared the SDIs of the total microbiota and the four dominant phyla in the prenatal fecal microbiota with the CB CCL17 levels (Table 4). The SDI of the total microbiota at 32 weeks of gestation was positively and significantly associated with CB CCL17 levels, while that of Firmicutes was negatively and significantly correlated with CB CCL17 levels. The SDI of Proteobacteria was also weakly and inversely correlated with CB CCL17 levels, although this correlation was not statistically significant.

### 3.5. No Relationship between CB CCL17 and Relative Abundances of Taxa in Maternal Feces

There were no statistically significant correlations between the CB CCL17 levels and the relative microbial abundances in maternal feces at either time point (Appendix A).

### 3.6. Maternal IgE Correlated with Proteobacteria Diversity in Maternal Feces during Pregnancy

Correlations between maternal non-specific IgE levels and the SDIs of the four dominant phyla in maternal feces and total microbiota were analysed at 32 weeks of gestation to assess the relationships with maternal allergic sensitization (Table 5). The SDI of Proteobacteria correlated positively and significantly with IgE levels, whereas the SDI of Firmicutes was weakly and inversely correlated with IgE levels, although this correlation was not statistically significant.

### 3.7. The Relative Abundances of Taxa in Maternal Feces during Pregnancy Were Not Correlated with Maternal Serum IgE

There were no statistically significant correlations between the non-specific IgE levels in maternal sera at 32 weeks of gestation and the relative microbial abundances in maternal feces (Appendix B).

## 4. Discussion

In addition to the role of the mother-baby vertical transmission of vaginal microbes at the moment of birth, our results suggest that maternal intestinal microbiota and SCFAs during pregnancy may influence the offspring’s fetal allergic immune response. Maternal SCFAs during pregnancy have also been shown to influence offsprings’ postnatal diseases other than allergic diseases. For instance, the sympathetic nerves, intestinal epithelium, and pancreas of embryos highly expressed the receptors to sense SCFAs originating from the maternal gut microbiota in mice [37]. The SCFA-receptor axes facilitate the development of neural cells, enteroendocrine cell, and pancreatic *β* cells, thereby shaping embryonic energy metabolism [37]. Moreover, the administration of SCFAs to rat breeders via drinking water prior to pregnancy and further treatment of the offspring with SCFAs after weaning let to ameliorate type 1 diabetes in rat offspring, whereas administration of SCFAs beginning at weaning did not protect from type 1 diabetes [38]. These findings suggested that maternal SCFAs during pregnancy play a key role in the regulation of disease susceptibility during postnatal life in the context of the developmental origins of health and disease theory, which is consistent with our findings.

Recent studies have evidenced that perturbations in intestinal microbiota in early infancy precede the development of atopic outcomes [39,40,41]. The caesarean delivery can instigate abnormal spectrum of gut microbiota in the newborn [42], the birth mode impact on the immune system development in infants born by C-section [43], and may account for the rising incidence of several serious health problems in children, including asthma and allergies [44]. We found an inverse association of CB CCL17 levels with the butyrate concentration in maternal feces at 12 weeks of gestation, and a positive association between CB CCL17 levels and each of the isobutyrate concentrations at 12 weeks of gestation and the valerate and the lactate concentrations at 32 weeks of gestation. Moreover, we also found that CB CCL17 levels were inversely associated with Firmicutes diversity, and positively associated with total microbiota diversity, at 32 weeks of gestation. These results show that the metabolites in maternal feces during pregnancy probably alter the fetal immune responses as well as the microbial composition.

The metabolites in maternal feces during pregnancy, notably butyrate, may be responsible for the lower CB CCL17 levels associated with higher Firmicutes diversity. The Firmicutes phylum includes the *Clostridium* genus. Seventeen strains within clusters of the Clostridia class in human feces, over half of which belonged to the *Clostridium* genus, have been found capable of producing butyrate, which drives cell differentiation and Treg cell expansion in the colon [34]. The butyrate concentration in maternal feces was negatively associated with both CB CCL17 and maternal IgE levels. Butyrate production by *Clostridium* species may cause increase in both maternal and fetal Treg cell numbers and modulate maternal IgE and CB CCL17 levels. However, we did not directly observe a correlation between butyrate concentration and either Firmicutes diversity or relative abundance in maternal feces. Allergy amelioration via maternal allergic status may partly explain why the Firmicutes diversity in the maternal intestine during pregnancy correlates with CB CCL17 levels. Further studies are required to establish the association between these factors.

Along with the SCFAs that are detrimental to CB CCL17 and maternal IgE levels, increased lactate concentrations may reflect the depression of lactate metabolism such that different end products such as butyrate are formed. The lactate-utilizing bacterial community plays a pivotal role in preventing lactate accumulation, which can have serious health effects [45]. Moreover, recent findings suggest that significant amounts of lactate may be converted into butyrate by dominant species in the human gut microbiota, such as *Eubacterium hallii*, *Anaerostipes caccae*, and *A**naerostipes coli* [46], which are all members of the Firmicutes phylum. Lactate accumulation may indicate the suppression of butyrate metabolism.

Moreover, McDonald et al. reported that valerate inhibited the growth of *Clostridium difficile*, both in vitro and in vivo [47]. The growth of Treg-inducible *Clostridium* species may also be inhibited by valerate. However, a more detailed analysis should be conducted to understand the mechanism whereby higher Firmicutes diversity contributes to the reduced CB CCL17 levels.

This study was subject to some limitations, as follows. The sample size was relatively small and limited to Japanese women. Our findings therefore require further investigation and validation by larger studies, in which better avoidance of type 1 errors would be possible. Moreover, the statistically necessary sample size was not determined prior to stool collection by power analysis, as the C-MACH is a pilot study. We were only able to investigate one Th2 chemokine; future studies incorporating another Th2 chemokine, such as CCL22, and a Th1 chemokine, such as CXCL10, may be better able to demonstrate the imbalance of the Th1 Th2 response in CB show stronger correlations with maternal total IgE serum levels.

## 5. Conclusions

We demonstrated that butyrate concentrations in maternal feces during pregnancy correlated negatively with CB CCL17 levels, whereas isobutyrate, valerate, and lactate concentrations correlated positively, suggesting that these metabolites play important roles in altering the offspring’s fetal immune responses. Moreover, total microbial diversity increased, while Firmicutes diversity decreased in maternal feces during pregnancy, while CB CCL17 levels rose. These results show that the composition of the maternal microbiota may influence the epithelial immune response in fetuses. Thus, we recommend that the maternal diet, in order to foster the development of a healthy microbiome during pregnancy, should include consumption of pro- and pre-biotic-containing foods, to reduce the risk of allergic diseases in the offspring.

## Figures and Tables

**Table 1 nutrients-13-02837-t001:** Demographic data (mean ± SD) of the study population.

	Participants of 12 Weeks of Gestation (*N* = 59)	Participants of 32 Weeks of Gestation (*N* = 58)
Maternal age (year)	33.09 (3.54) (unknown *n* = 2)	33.43 (3.44) (unknown *n* = 2)
Pregravid BMI	21.07 (2.99) (unknown *n* = 2)	20.98 (2.89) (unknown *n* = 2)
Sex of neonates (male/female)	21/33 (unknown *n* = 5)	25/32 (unknown *n* = 1)
Delivery mode (spontaneous/caesarean)	49/5 (unknown *n* = 5)	50/7 (unknown *n* = 1)
Birth height (cm)	49.40 (1.44) (unknown *n* = 5)	49.33 (1.50) (unknown *n* = 1)
Birth weight (g)	3038.42 (283.16) (unknown *n* = 5)	3045.61 (306.26) (unknown *n* = 1)
Head circumference (cm)	33.22 (1.04) (unknown *n* = 5)	33.22 (1.12) (unknown *n* = 1)
Gestational age (week)	38.94 (1.02) (unknown *n* = 5)	38.95 (0.99) (unknown *n* = 1)
CB CCL17 level (100 pg/mL)	10.38 (3.99) (unknown *n* = 5)	10.67 (4.09) (unknown *n* = 3)
Maternal total IgE level at 32 weeks of gestation (IU/mL)	105.08 (105.15) (unknown *n* = 5)	116.18 (123.73) (unknown *n* = 1)

Abbreviations: BMI, body mass index; CB, cord blood; CCL17, chemokine (C-C motif) ligand; IgE, immunoglobulin E; SD, standard deviation.

**Table 2 nutrients-13-02837-t002:** Associations between CB CCL17 and short-chain fatty acid levels in maternal feces ^1^.

			95% CI	
	*β*	*P*	Lower	Upper
12 weeks of gestation (*n* = 53)				
Acetate	0.00	1.00	0.00	0.00
Propionate	−2.05	0.85	−22.81	18.70
Isobutyrate	305.58	0.03 *	36.55	574.61
Butyrate	−57.42	0.04 *	−112.96	1.88
Isovalerate	0.00	1.00	0.00	0.00
Valerate	7.45	0.96	−311.56	326.46
Hexanoate	0.00	1.00	0.00	0.00
Lactate	0.00	1.00	0.00	0.00
Succinate	0.00	1.00	0.00	0.00
32 weeks of gestation (*n* = 54)				
Acetate	0.00	1.00	0.00	0.00
Propionate	0.00	1.00	0.00	0.00
Isobutyrate	20.37	0.95	−564.07	604.82
Butyrate	−38.72	0.18	−95.84	18.40
Isovalerate	−289.32	0.56	−1274.34	695.69
Valerate	440.27	0.003 **	152.86	727.68
Hexanoate	0.00	1.00	0.00	0.00
Lactate	184.48	0.01 *	43.65	325.32
Succinate	−2.60	0.52	−10.51	5.32

^1^ Parameters were calculated using adaptive LASSO. Statistically significant (*p* ≤ 0.05) values are indicated by * and strongly significant (*p* ≤ 0.01) values by **. CB CCL17 levels were converted to their natural logarithms prior to analysis. Abbreviations: CB, cord blood; CCL17, chemokine (C-C motif) ligand; CI, confidence interval; LASSO, least-absolute shrinkage and selection operator. A mother who used antibiotics during pregnancy was excluded from the analysis.

**Table 3 nutrients-13-02837-t003:** Associations between maternal non-specific serum IgE levels at 32 weeks of gestation and short-chain fatty acid levels in maternal feces ^1^.

			95% CI	
	*β*	*P*	Lower	Upper
32 weeks of gestation (*n* = 56)				
Acetate	4.58	0.25	−3.29	12.44
Propionate	8.01	0.11	−1.76	17.78
Isobutyrate	0.00	1.00	0.00	0.00
Butyrate	−31.72	0.04 *	−62.08	−1.36
Isovalerate	0.00	1.00	0.00	0.00
Valerate	0.00	1.00	0.00	0.00
Hexanoate	−104.10	0.52	−418.78	210.58
Lactate	28.52	0.03 *	2.27	54.78
Succinate	0.00	1.00	0.00	0.00

^1^ Parameters were calculated using adaptive LASSO. Statistically significant (*p* ≤ 0.05) values are indicated by *. Non-specific IgE levels were converted to their natural logarithms prior to analysis. Abbreviations: CI, confidence interval; LASSO, least-absolute shrinkage and selection operator; IgE, immunoglobulin E. A mother who used antibiotics during pregnancy was excluded from the analysis.

**Table 4 nutrients-13-02837-t004:** Associations between the SDIs of total microbiota and four phyla in maternal feces and CB CCL17 levels ^1^.

			95% CI	
Phylum	*β*	*p*	Lower	Upper
12 weeks of gestation (*n* = 53)				
Total microbiota	−0.18	0.35	−0.57	0.21
Firmicutes	0.19	0.36	−0.23	0.62
Proteobacteria	−0.10	0.51	−0.41	0.21
Actinobacteria	0.03	0.87	−0.31	0.36
Bacteroidetes	0.05	0.72	−0.24	0.35
32 weeks of gestation (*n* = 54)				
Total microbiota	0.95	0.01 *	0.26	1.64
Firmicutes	−0.69	0.01 *	−1.22	−0.17
Proteobacteria	−0.28	0.05	−0.56	0.002
Actinobacteria	−0.17	0.26	−0.48	0.13
Bacteroidetes	−0.01	0.95	−0.29	0.27

^1^ All parameters were calculated using multivariable regression analysis. Statistically significant (*p* ≤ 0.05) values are indicated by *. CB CCL17 levels were converted to their natural logarithms prior to analysis. Abbreviations: CB, cord blood; CCL17, chemokine (C-C motif) ligand 17; CI, confidence interval; SDI, Shannon diversity index. A mother who used antibiotics during pregnancy was excluded from the analysis.

**Table 5 nutrients-13-02837-t005:** Associations between the SDIs of total microbiota and four phyla in maternal feces at 32 weeks of gestation and non-specific IgE levels in maternal sera ^1^.

			95% CI	
Phylum	*β*	*P*	Lower	Upper
32 weeks of gestation (*n =* 56)				
Total microbiota	1.24	0.32	−1.23	3.70
Firmicutes	−1.76	0.07	−3.68	0.16
Proteobacteria	1.06	0.04 *	0.04	2.08
Actinobacteria	−0.72	0.19	−1.83	0.38
Bacteroidetes	−0.46	0.35	−1.43	0.51

^1^ All parameters were calculated using multivariable regression analysis. Statistically significant (*p* ≤ 0.05) values are indicated by *. Non-specific IgE levels were converted to their natural logarithms prior to analysis. Abbreviations: CI, confidence interval; IgE, immunoglobulin E; SDI, Shannon diversity index. A mother who used antibiotics during pregnancy was excluded from the analysis.

## Data Availability

The data that support the findings of this study are available from the C-MACH research committee upon reasonable request with permission; however, restrictions apply to the availability of these data, which were used under license for the current study, and thus are not publicly available at present.

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
