# Peer review of "Association of the Maternal Gut Microbiota/Metabolome with Cord Blood CCL17"

_nutrients, 2021, doi:10.3390/nu13082837_

Round 1
Reviewer 1 Report
In this work, the authors have studied the associations between CB CCL17 and gut SCFA levels in healthy pregnant Japanese women. CB CCL17 serum levels at birth, and maternal non-specific IgE levels in maternal sera at 32 weeks of gestation were measured. Maternal stool samples were collected at 12 (n = 59) and 32 (n = 58) weeks of gestation for gut microbiota analysis, based on barcoded 16S rRNA sequencing and metabolite levels.
The study is very interestingly, howover some aspect must be treated in materials and methods. Particularly, the analysis of outpout of NGS such as the output from primary analysis serves as the starting point for NGS data analysis for various instrument-specific pipelines and the authour should improve the description of these aspects.
Moreover, the discussion seems very simple and poor and the authors should be improved the comments also in relation to the recent literature.
However, I think that it can stimulate the scientific debate and drive further research in the near future. And I think the authors should consider the following points and the text will be emended accordingly. Indeed, I recommend the authors the deeply consider all the following suggestions, which are a genuine attempt to help them to improve the manuscript quality.
Author Response
We appreciate the time and effort you have dedicated to providing insightful feedback. Accordingly, we have revised our manuscript in the suggested ways, which we believe have strengthened it.
The study is very interestingly, howover some aspect must be treated in materials and methods. Particularly, the analysis of outpout of NGS such as the output from primary analysis serves as the starting point for NGS data analysis for various instrument-specific pipelines and the authour should improve the description of these aspects.
We thank you for your comment. We have added the following sentences to address this issue (page 3, line 119-132):
“Dual indexes and Illumina sequencing adapters were attached to PCR products using the Nextera XT Index Kit (Illumina). After purification of the amplicon using AMPure XP beads, these samples were quantified using a Quant-iT PicoGreen ds DNA Assay Kit (Life Technologies Japan, Tokyo, Japan). Mixed samples were prepared by pooling approximately equal amounts of PCR amplicons from each sample. A pooled sample library with 20% denatured PhiX spike-in (SeqMatic LLC, Fremont, CA, USA) was sequenced on the Miseq using a 500 cycle kit (Illumina).
Taxonomic assignments and estimation of relative abundance of sequencing data were performed using the analysis pipeline of the QIIME software package.[1] Chimera checking was performed using UCHIME.[2] An operational taxonomic unit (OTU) was defined at 97% similarity. The OTU was assigned a taxonomy based on a comparison with the Greengenes database using RDP classifier.[3, 4] The proportions of identified taxa in each sample were summarized and the amount of bacterial diversity was calculated using vegan package in R software.”
Moreover, the discussion seems very simple and poor and the authors should be improved the comments also in relation to the recent literature.
According to your suggestion, we have added the following sentences (page 8, line 271-283) and cited Kimura I. et al. 2020 (:[37]) and Needell J. et al. 2017(:[38]):
“Maternal SCFAs during pregnancy have also been shown to influence offsprings’ postnatal diseases other than allergic diseases. For instance, the sympathetic nerves, intestinal epithelium, and pancreas of embryos highly expressed the receptors to sense SCFAs originating from the maternal gut microbiota in mice [37]. The SCFA-receptor axes facilitate the development of neural cells, enteroendocrine cell, and pancreatic βcells, thereby shaping embryonic energy metabolism [37]. Moreover, the administration of SCFAs to rat breeders via drinking water prior to pregnancy and further treatment of the offspring with SCFAs after weaning let to ameliorate type 1 diabetes in rat offspring, whereas administration of SCFAs beginning at weaning did not protect from type 1 diabetes[38]. These findings suggested that maternal SCFAs play a key role in the regulation of disease susceptibility during postnatal life in the context of the developmental origins of health and disease theory, which is consistent with our findings.”
We thank you for your comment.

Reviewer 2 Report
I've read with attention the paper of Hiromi Tanabe et al. In this manuscript examined the link between maternal gut microbiota, SCFAs, and CB CCL17 in healthy pregnant Japanese women. The experimental setting of the manuscript is interesting. The manuscript is logically organized and well-written in English. The background and aim of the study have been clearly defined. The methodology applied is correct and sufficiently described, the results are reliable and clearly presented. The discussion is clear and centered on other relative studies. I have only one small suggestion. I don’t have an objection to write like this but I suggest to avoid long sentences (like lines 262-268) as it can make reading difficult.
Author Response
Responses to the reviewers
Reviewer 2
We appreciate the time and effort you and the reviewers have dedicated to providing insightful feedback. Accordingly, we have revised our manuscript in the suggested ways, which we believe have strengthened it.
I've read with attention the paper of Hiromi Tanabe et al. In this manuscript examined the link between maternal gut microbiota, SCFAs, and CB CCL17 in healthy pregnant Japanese women. The experimental setting of the manuscript is interesting. The manuscript is logically organized and well-written in English. The background and aim of the study have been clearly defined. The methodology applied is correct and sufficiently described, the results are reliable and clearly presented. The discussion is clear and centered on other relative studies. I have only one small suggestion. I don’t have an objection to write like this but I suggest to avoid long sentences (like lines 262-268) as it can make reading difficult.
Indeed, the sentence was too long and unintelligible. We divided it into two sentences as follows (page 8, lines 284-289):
“Recent studies have evidenced that perturbations in intestinal microbiota in early infancy precede the development of atopic outcomes[39-41]. The caesarean deliverly can instigate abnormal spectrum of gut microbiota in the newborn[42], the birth mode impact on the immune system development in infants born by C-section[43], and may account for the rising incidence of several serious health problems in children, including asthma, allergies[44]. ”
We thank you for your comment.

Round 2
Reviewer 1 Report
The changes were made the work can be accepted.